# Model-based analysis of the acute effects of transcutaneous magnetic spinal cord stimulation on micturition after spinal cord injury in humans

**Mahshid Fardadi**[1]*, **J. C. Leiter**[2], **Daniel C. Lu**[3], **Tetsuya Iwasaki**[1]

**1** Department of Mechanical Engineering, University of California, Los Angeles, California, United States of America, **2** White River Junction VA Medical Center, White River Junction, Vermont, United States of America, **3** Department of Neurosurgery, University of California, Los Angeles, California, United States of America

\* mfardadi@ucla.edu

**Data Availability Statement:** Code for data cleaning and analysis is provided as part of the replication package. It is available at https://github.com/mfardadi/Model-based-Analysis.

## Abstract

### Aim

After spinal cord injuries (SCIs), patients may develop either detrusor-sphincter dyssynergia (DSD) or urinary incontinence, depending on the level of the spinal injury. DSD and incontinence reflect the loss of coordinated neural control among the detrusor muscle, which increases bladder pressure to facilitate urination, and urethral sphincters and pelvic floor muscles, which control the bladder outlet to restrict or permit bladder emptying. Transcutaneous magnetic stimulation (TMS) applied to the spinal cord after SCI reduced DSD and incontinence. We defined, within a mathematical model, the minimum neuronal elements necessary to replicate neurogenic dysfunction of the bladder after a SCI and incorporated into this model the minimum additional neurophysiological features sufficient to replicate the improvements in bladder function associated with lumbar TMS of the spine in patients with SCI.

### Methods

We created a computational model of the neural circuit of micturition based on Hodgkin-Huxley equations that replicated normal bladder function. We added interneurons and increased network complexity to reproduce dysfunctional micturition after SCI, and we increased the density and complexity of interactions of both inhibitory and excitatory lumbar spinal interneurons responsive to TMS to provide a more diverse set of spinal responses to intrinsic and extrinsic activation of spinal interneurons that remains after SCI.

### Results

The model reproduced the re-emergence of a spinal voiding reflex after SCI. When we investigated the effect of monophasic and biphasic TMS at two frequencies applied at or below T10, the model replicated the improved coordination between detrusor and external

**Funding:** This material is the result of work supported by a University of California Chancellor's Postdoctoral Fellowship and from the Louis and Harold Price Foundation (MF), H & H Evergreen Foundation, Jonathan and Susan Dolgen Foundation, and Department of Defense (DOD) research grant SC103209. The funders had no role in study design, data collection and analysis, decision to publish, or preparation of the manuscript. None of the authors received salary from any of the funders.

**Competing interests:** The authors have declared that no competing interests exist.

urethral sphincter activity that has been observed clinically: low-frequency TMS (1 Hz) within the model normalized control of voiding after SCI, whereas high-frequency TMS (30 Hz) enhanced urine storage.

## Conclusion

Neuroplasticity and increased complexity of interactions among lumbar interneurons, beyond what is necessary to simulate normal bladder function, must be present in order to replicate the effects of SCI on control of micturition, and both neuronal and network modifications of lumbar interneurons are essential to understand the mechanisms whereby TMS reduced bladder dysfunction after SCI.

### Author summary

We developed a computer model using Hodgkin-Huxley representations of spinal interneurons to simulate normal bladder control, dysfunctional control of micturition after SCI, and clinically observed bladder responses to transcutaneous magnetic stimulation (TMS) of the lumbar spine. We incorporated an infantile spinal voiding reflex by adding lumbar interneurons to make a spinal voiding reflex possible and modified descending control of these neurons to inhibit the infantile reflex in normal adults. After SCI, the descending inhibition of the spinal voiding reflex is lost, and the spinal voiding reflex re-emerges. To make the lumbar spine response to TMS, it was necessary to add two excitatory interneurons and one inhibitory neuron in the lumbar spine: these neurons manifested frequency-specific responses to TMS. Based on the dynamical properties of the inhibitory and excitatory neurons responding to TMS, low frequency TMS (1 Hz) permitted the action of the re-emergent spinal voiding reflex to coordinate bladder contraction and urethral relaxation so that effective bladder emptying was re-established during TMS. Higher frequency TMS contracted the urethral sphincter and prevented simulated leaking and incontinence. Greater lumbar interneuronal complexity, consistent with animal experiments, is required to simulate bladder responses to SCI and TMS.

## 1. Introduction

Many patients with spinal cord injury (SCI) above the lumbar spine lose voluntary control of bladder function, either partially or entirely, and develop a neurogenic bladder, which manifests as detrusor-sphincter dyssynergia (DSD)—a lack of coordination between the detrusor and sphincter muscles that prevents effective bladder emptying. On the other hand, patients with SCI at the lumbosacral spinal level have an areflexic, flaccid bladder, reduced bladder capacity, and an inability to store urine leading to incontinence [1]. Electromagnetic stimulation of the spinal cord is a promising approach to restore lower urinary tract (LUT) functions after SCI. Epidural electrical stimulation and transcutaneous magnetic stimulation (TMS) of the lumbar spine below the level of injury influence neural activities within the spine and have the potential to foster recovery of bladder function after SCI [2–5]. The mechanisms whereby spinal cord stimulation affects the activities of the neural circuits in the lumbosacral spine governing LUT function and improve control of micturition remain uncertain.

Urine storage and release are maintained by a combination of higher cortical mechanisms and semi-autonomous functions embedded in the brainstem and in spinal reflex pathways in normal individuals [6–11]. Mathematical modeling of micturition is a useful analytical process that compels investigators to define explicitly, in mathematical terms, the physiological processes controlling urine storage and release. Previous investigators modelled the biomechanical features of the bladder with limited external neurophysiological controls or focused on the neural elements controlling the bladder (and largely ignored the biomechanism constraints of the bladder) [8–11]. In either case, the models were based on mechanistic descriptions of physiological processes that incorporated the smallest set of neural or biochemical features needed to describe the behavior being modelled, and the models described control of the LUT in healthy subjects. Muscle dynamics have been described by Hill equations (which describe the activation of muscles controlling the bladder in terms of muscle length, tension, and velocity of shortening), but neural elements were represented by static models, such as the weighted sum of inputs operating around a threshold or a logical on/off switch [10]. More recently, a model including the supraspinal voiding reflex was used to examine the effect of pudendal nerve stimulation on bladder contraction [12]. The neural elements were modeled as a network of integrate-and-fire neurons, and nerve stimulation was simulated as action potentials in the pudendal nerve. Last, a model based on what might be described as weighted logic gates to simulate excitatory or inhibitory neurons within the circuit of micturition control produced on/off reflex responses that simulated normal urine storage and micturition [13].

None of these previous models addressed the effect of SCI on micturition. Given the limited capacity to perform invasive measurements in humans, modeling the dynamical behavior of the neural circuits controlling micturition provides a useful mechanism to explore how SCI may alter control of micturition and how spinal cord stimulation may ameliorate control of bladder function. Therefore, we developed a mathematical model of the spinal micturition circuit as a dynamical network of neural nodes described by the Hodgkin-Huxley equations so that spinal cord stimulation was incorporated directly as a transient opening of the sodium channels. The activity of the network controlling micturition was dictated by the nature of afferent inputs, the connectivity of the network nodes, and the synaptic and ionic characteristics of the model neurons [14]. The basic architecture of a simplified network model of micturition was simulated based on an established model of normal storage/voiding functions as controlled by the spinal storage reflex and supraspinal control of voiding [6]. Once the simplified model controlling micturition under normal, uninjured circumstances was constructed, we introduced the minimum number of additional, novel features needed to create a single, unified model with a fixed network architecture that reproduced observations both before and after SCI and with and without TMS applied to the lumbar spine.

## 2. Methods

Models developed previously, which sufficed to model the main features of micturition in normal humans, will be referred to as the 'traditional' model. Additions and modifications to the traditional model, which were necessary to simulate dysfunctional micturition following SCI and to make the lumbar spine responsive to TMS, will be referred to as the 'enhanced' model.

### 2.1 The Traditional model of healthy control of micturition

Several previous models embody the hierarchical structure of control of urination in normal subjects [6,13,15–17], and though they differ slightly from each other, they share a core architecture describing the neural feedback controlling micturition [15,18]. The features of this traditional model are shown schematically in Fig 1A [15,18].

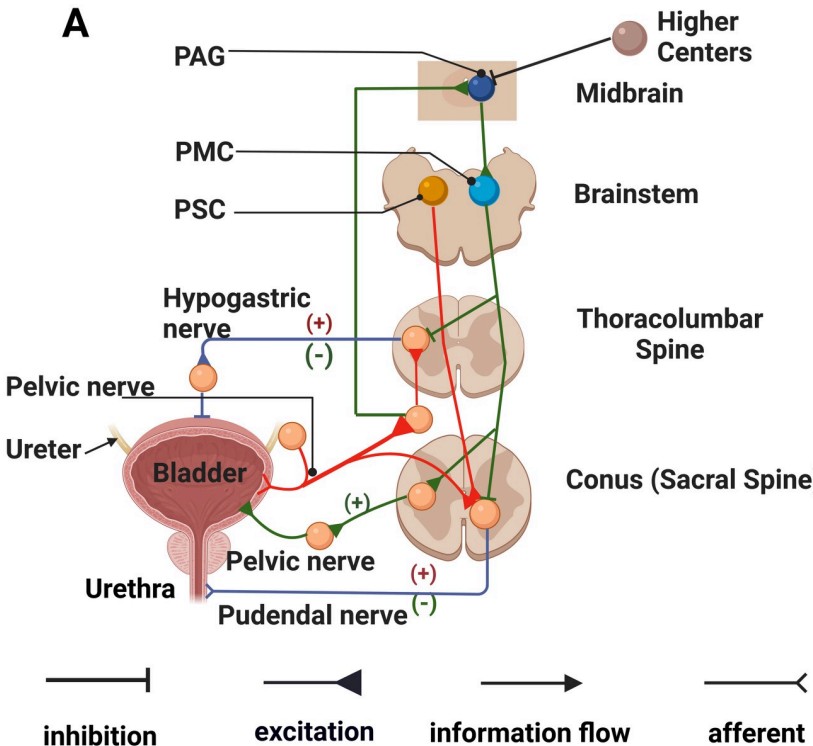

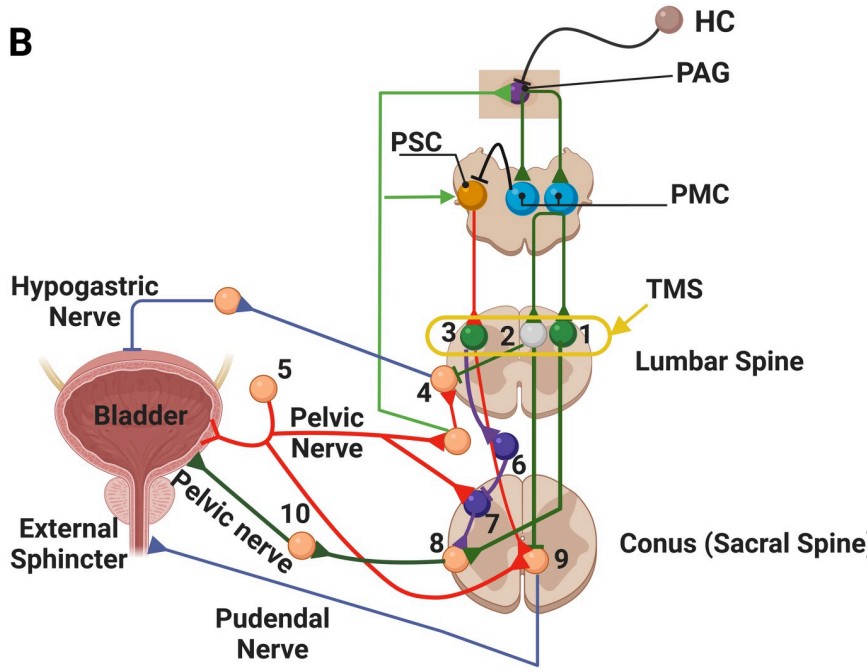

**Fig 1.** The traditional (A) and enhanced (B) neural circuits of micturition have been adapted and redrawn from Fig 5 in Fowler et al. [18]. The circuits for both micturition and continence, shown separately by Fowler et al [18], were incorporated into a single circuit in Fig 1A and 1B. The circuit elements controlling urine storage are shown in red; the circuit elements controlling voiding are shown in green; and shared parts of the network are shown in blue. Neuronal populations are represented by individual nodes in the model calculations (and in the original figure drawn by Fowler et al. [18]). The figure emphasizes efferent control of the bladder and urethra, but the nerves innervating the bladder

and urethra also carry afferent information, which is not, with the exception of the pelvic nerve, explicitly incorporated into either the traditional or enhanced models. The afferent connection of the pelvic nerve to the bladder is shown in red and the efferent connection is shown separately in green for clarity—even though the afferent and efferent activity run in a single nerve in reality. Only the efferent activity in the pudendal nerve controlling contraction of the external sphincter is shown and incorporated in the model (even though the pudendal nerve carries afferent information that contributes to bladder control). The circuits shown in Fig 1A, are adequate to explain the storage and voiding operations of healthy individuals. When an upper-level injury occurs at T10 or above (green dashed line, Fig 1B), the communication of the spinal circuits with higher centers was blocked. Additions to the traditional model were necessary to allow the spinal, interneuronal circuit controlling micturition to replicate both overactive bladder and detrusor-sphincter dyssynergia, which are seen after SCI. Three additional elements have been added to the circuit shown in B in order to replicate dysfunctional micturition after SCI. We added a spinal voiding reflex loop involving the afferent pelvic nerve (Node 7) that communicates with preganglionic elements in the spine that control detrusor contraction (Node 8) (shown in yellow; note also that an inhibitory Node 6 was added to the model to allow the pontine storage center to suppress the spinal voiding reflex in normal subjects). Afferent information flow through the pelvic nerve to the PSC (to enhance storage) was made explicit (green arrow), and last, an inhibitory connection from the PMC to the PSC was added so that storage was inhibited during the act of micturition (shown in black). In addition, three lumbar interneuronal nodes (excitatory Nodes 1 and 3 –green, and one inhibitory Node 2—grey) were added to receive TMS. PAG—periaqueductal grey; PSC—pontine storage center; PMC—pontine micturition center; HC—higher cortical centers; SCI—spinal cord injury; TMS—transcutaneous magnetic stimulation; Nodes 1 and 3 -excitatory interneurons; Node 2 inhibitory interneuron; Node 4 –efferent hypogastric nerve ganglion; Node 5—afferent pelvic nerve ganglion; Node 6 lumbar inhibitory interneuron; Node 7 a lumbar interneuron; Node 8 preganglionic parasympathetic neuron; Node 9 somatic motor neuron. The unnumbered nodes were maintained for continuity with the original figure, but these nodes were not incorporated in the model since they incorporate no internodal information processing (they simply pass information to the next node without modification as explained in the text). The colors of the nodes differ to enhance visual contrast, but the colors have no mechanistic significance. This figure was created in part with BioRender.com.

### 2.1.1 Urine storage.

Higher centers in the brain control volitional aspects of micturition [7,18,19], and brainstem nuclei regulate spinal reflexes that provide semi-autonomous control of urine storage and micturition [13,15,18]. Just beneath the higher centers controlling urination, the mechanisms facilitating urine storage in the bladder are organized under the pontine storage center (PSC), and the control of micturition is organized under the pontine micturition center (PMC).

During urine storage as shown in Fig 1A, bladder pressure increases as the bladder fills, and pelvic nerve afferents convey this volume-related information to the lumbosacral spinal cord, where this and other afferent information from the pelvic floor and urethra are communicated to second-order neurons, which in turn communicate information about the state of the bladder through spinal interneurons to the periaqueductal gray (PAG) and PMC [20–25]. More directly within the spinal cord, bladder-related information is communicated to sympathetic preganglionic neurons in the intermediolateral cell column between T10 and L2. Preganglionic sympathetic activity is directed to the inferior hypogastric ganglion, and postganglionic sympathetic activity in the hypogastric nerve relaxes the detrusor so that the bladder remains flaccid and can accommodate increasing urine volumes during urine storage [20–26]. This bladder-derived afferent information provides a 'tonic influence' to activate the PSC, which in turn stimulates neurons in the Nucleus of Onuf that contract muscles of the pelvic floor and the external urethral sphincter [18,24,27,28].

### 2.1.2 Micturition.

Micturition and bladder emptying rely on a combination of inhibitory and excitatory interactions to contract the bladder and relax the urethral sphincter so that urine is expelled through the urethra [15,18,24]. The central organization of the micturition reflex derives from ascending projections from the lumbosacral cord that transmit information regarding bladder filling to the periaqueductal gray (PAG) in the brainstem, as noted above [21,24,29]. Once the frequency of afferent pelvic nerve activity surpasses a threshold detected within the PAG, indicating that the bladder is full, the PAG relays this information to the PMC and higher brain centers that integrate bladder fullness with other cues to determine if

urination is physiologically necessary. If urination is necessary and socially appropriate, volitional urination is initiated, during which tonic activation of the PSC is likely withdrawn, and the pontine micturition center (PMC) becomes active. This reciprocal control of PMC and PSC activity is likely effected by higher centers in the midbrain and cortex. When activated, the PMC projects to the lumbosacral cord to execute voiding during which detrusor contraction is activated; sympathetic efferent activity (relaxing the detrusor muscle) is inhibited; and contraction of the urethral sphincter and muscles of the pelvic floor is actively inhibited by GABAergic mechanisms [8,23–25,28]. After bladder emptying, the storage cycle (detrusor inhibition and urethral sphincter contraction) resumes under control of the PSC.

## 2.2 No detrusor-sphincter dyssynergia or bladder overactivity within the traditional model

Spinal cord injury above the lumbar level blocks or weakens supraspinal control of voiding and prevents reciprocal coordination between the activity of the bladder and the urethra. Previous models of 'normal' bladder and urethral sphincter control have not replicated the usual patterns of bladder dysfunction observed after a spinal cord injury, such as an overactive bladder (disinhibited detrusor contractions and urine leakage) or detrusor-sphincter dyssynergia (simultaneous contraction of the detrusor and the urethral sphincter). The dysfunctional patterns of bladder activity after a SCI derive from interactions among spinal networks of interneurons, neuronal plasticity, and reorganization of spinal circuitry that foster emergence of a latent spinal voiding reflex [1,6,30–33].

When a SCI occurs at T10 or above (an upper-level injury; the red dashed line in Fig 1B), communication between the brainstem circuits and the spinal micturition circuits is blocked or attenuated. The control of urine storage in the bladder remains intact, as autonomous feedback below the level of injury enables inhibition of bladder contraction and excitation of the urethral sphincter However, bladder emptying is dysfunction: the lack of descending signals from the PMC allows detrusor-sphincter dyssynergia to emerge, and, depending on the level of SCI, the bladder may not simply be overfilled; it can also be overactive, which may result in incontinence.

## 2.3 Proposed model enhancements to LUT control following SCI and during TMS

We addressed the failure of the traditional model to replicate the LUT dysfunction seen after spinal cord injury at T10 or above in two steps: we introduced interneuronal elements and patterns of neuronal connectivity to generate a infantile spinal voiding reflex, which has been demonstrated in previous studies of neonatal animals [1,32,34]. Second, though the circuitry for the spinal voiding is present in adults, the reflex is inactive. Therefore, we introduced descending pontine inhibitory control over the spinal voiding reflex so that the reflex circuitry is present, but the reflex activity remains latent in intact, adult animals and humans. In addition, we added lumbar interneurons responsive to TMS to provide a mechanism whereby TMS may modify lumbar spinal reflexes regulating urinary behaviors. These changes are aligned with physiological studies and proposed mechanisms of responses to SCI and TMS [2,32]. Last, we made some changes to make the model computationally tractable. The additions to the traditional model that we made to accommodate SCI and the response to TMS are summarized in Table 1.

**2.3.1 Implementation of a spinal voiding reflex.** Bladder function is not stable after an upper-level spinal cord injury, and threshold-based volume-emptying can emerge after a SCI. Therefore, we incorporated an infantile spinal voiding reflex into the model. This reflex is

**Table 1. Structural additions to the traditional model that were necessary to replicate bladder dysfunction after SCI and replicate the response to lumbar TMS.**

| Function added | Structural model elements added/modified to implement the new function | |
|---|---|---|
| Infantile spinal voiding reflex | A lumbar interneuronal node added to implement threshold voiding (node 7) | |
| | Separate excitatory and inhibitory drive from the PMC (two nodes added) | |
| | Inhibitory synaptic connection to lumbar interneuron (Node 7) controlling infantile voiding reflex (to keep reflex latent in normal subjects) from PMC (Node2). | |
| | After SCI: | Severed connections of PMC to lumbar interneurons, |
| | | Increased excitatory synaptic weights:<br>    Nodes 6 and 7<br>    Nodes 5 and 7<br>Increased inhibitory synaptic strength:<br>    Node 2 to Node 4<br>Decreased excitatory synaptic strength:<br>    Node 5 and sympathetic preganglionic neurons that communicate with Node 4. |
| Responsiveness to lumbar TMS | Incorporated a local network of excitatory and inhibitory interneurons in the lumbar spine (Nodes 1, 2, and 3). | |
| | Incorporated separate inhibitory and excitatory neurons responsive to TMS. | |
| Computational simplifications: | Inserted direct communication of bladder-related information via the pelvic nerve to the PSC (green arrow) and direct inhibition of the PSC by the PMC when it is excited (direct black inhibitory connection from PMC to PSC). | |
| | Simplified by not incorporating pudendal afferent information. | |
| | When a node passed information without any internodal modification, the neuronal dynamics of the node were not included in the model—information was passed directly to the effector node with addition of an appropriate time constant (unnumbered nodes). | |

resident in the lumbar spine and is thought to be suppressed in adults, but emerges after SCI through neural reorganization [1,18,31,34]. To incorporate a spinal voiding reflex into the model, we added more complex interneuronal interactions in the lumbar spine and modified the synaptic strength between selected interneurons below the level of spinal injury. These changes are physiologically plausible and represent the range of plastic mechanisms that exist in the spine in paraplegic animals [1,32].

The afferent loop of the spinal voiding reflex is carried by the pelvic nerves (red fibers, Node 5, in Fig 1B and already in the traditional model), which activates a new interneuron indicated by the red connection to Node 7. The efferent control of the spinal voiding reflex is mediated by the intraspinal elements within the lumbar spine from Node 7 to the parasympathetic, preganglionic intraspinal, Node 8, which when activated, elicits parasympathetic detrusor contraction. Afferent pelvic nerve activity also activates sympathetic, preganglionic neurons in the intermediolateral column of the thoracolumbar spine that elicit sympathetic suppression of detrusor contraction (Node 4), and pelvic efferent signaling activates the external sphincter (Node 9) [6,32,33]. The unnumbered nodes were in the traditional model (as drawn by Fowler et al [18]), but information passes through these nodes unaltered by internodal interactions, and these unnumbered nodes were not explicitly included in the model.

**2.3.2 Inhibition of the spinal voiding reflex.** Once the neural circuit of supraspinal, volitional control of micturition is fully developed in adults, the spinal voiding reflex must be inhibited [32], likely by descending signals from PSC. Therefore, we simulated a polysynaptic, spino-bulbo-spinal reflex process based on the activation of the pontine storage center (PSC) by ascending afferent information (indicated in green extrinsic to the spine in Fig 1B) to inhibit the infantile spinal voiding reflex in healthy adults by descending activation of Node 6,

which inhibits Node 7 (to keep the infantile voiding reflex latent in adults) and prevents uncontrolled spontaneous voiding once bladder filling reaches a threshold—the typical pattern of voiding in infants [18,31,34]. Simply adding nodes within the lumbar spine was insufficient to generate a functioning spinal voiding reflex since the reflex normally regresses as infants mature. The emergence of the reflex after SCI involves elements of neuroplasticity [6,13,32,35]. Using neuroplastic process similar to those proposed by de Groat et al. [32], we modeled neuroplasticity after SCI by increasing the synaptic weights of the connection between Nodes 6 and 7, Nodes 5 and 7, and increased the strength of inhibition by Node 2 of sympathetic preganglionic neurons represented by Node 4, and we decreased the synaptic weight of the excitatory connection between Node 5 and sympathetic preganglionic neurons that communicate with Node 4.

**2.3.3 Making the model responsive to lumbar TMS.** To permit the lumbar spine to respond to TMS, we incorporated a local network of excitatory and inhibitory interneurons in the lumbar spine (Nodes 1, 2, and 3 in Fig 1B) that relay the descending signals from PSC and PMC to the lumbosacral motor neurons driving the detrusor and sphincter muscles in normal circumstances. These interneurons were the recipients of excitation by TMS below the level of SCI that then modified LUT responses mediated by spinal interneurons within the enhanced model of the control of micturition. We need three interneurons to allow TMS to alter descending excitatory influences of the PSC and both inhibitory and excitatory influences from the PMC.

**2.3.4 Computational simplifications.** We made some simplifications to the model to reduce the computational time and complexity of executing the model. These simplifications did not change the fundamental interactions among lumbosacral spinal interneurons in the network controlling micturition, they did make model execution run more smoothly.

We represented the PMC by two nodes (shown in blue and labelled PMC in Fig 1B) one of which inhibits the PSC so that voiding and storage are reciprocally inhibited. Although this is shown as direct inhibition in the model, connections between the PMC and the PSC are sparse [28,36], and this inhibition of urine storage during micturition is probably effected by withdrawal of excitatory inputs to the PSC and removal of descending inhibition of the PMC from higher centers in the brain that determine the timing and place of volitional urination. The other PMC Node has the same function as it did in the traditional model: to excite detrusor contraction and inhibit urethral sphincter contraction. However, it was necessary to represent the PMC by two nodes to allow both inhibitory and excitatory descending control of micturition from the PSC and PMC.

Although the PAG has a tonic influence on the PSC, the exact afferent pathway conducting information about the state of the system to the PSC is not well defined in that spinal fibers conducting afferent information from the bladder and pelvic floor do not lead directly to the PSC [18]. The flow of information communicating the state of the bladder and urethra to the PSC is polysynaptic and not direct as there are few to no anatomical connections between the PMC and PSC [25,28]. Retrograde tracers injected into the upper sacral region label neurons in the PSC [24], and the activity of neurons within the PSC appears to track the state of bladder filing—indicating that afferent information from the bladder reaches the PSC [37]. To indicate that afferent information influences PSC activity indirectly, we used a green arrow from the pelvic nerve to the PSC drawn outside the spine (Fig 1B). Both the more direct communication of bladder-related information via the pelvic nerve to the PSC (green arrow) and direct inhibition of the PSC by the PMC when it is excited (direct black inhibitory connection from PMC to PSC in Fig 1B) were necessary to effect smooth switching between urine storage and micturition.

The traditional circuit, as shown in Fig 1A, is simplified in that pelvic floor and sphincter afferents in the pudendal nerve are not considered—though pudendal afferent information related to urine flow in the urethra that is transmitted through the spine to the periaqueductal grey (PAG), helps coordinate reciprocal detrusor-sphincter activation/inhibition during micturition and urine storage [38–40].

## 2.4 Model implementation

Each node in the network, shown in Fig 1B, was modeled as a single Hodgkin-Huxley neuron connected to other neurons in the network as indicated in the figure. The behavior of each node (representing a population of neurons) was described by Hodgkin-Huxley type equations:

$$c\dot{v} + (g_{Na} + g_{TMS})(v - v_{Na}) - g_K(v - v_K) - g_L(v - v_L) + g_{syn}(v - v_{syn}) = 0 \tag{1}$$

where v is the membrane potential (the dot above the v denotes the time derivative), c is the cell membrane capacitance, and each term of the form $g(v–v_r)$ is the channel current through conductance g due to the voltage difference between v and the reversal potential $v_r$ (the details are provided in the S1 Materials). Each node was modeled as a single-compartment model, except the pelvic nerve, which was described as a multi-compartment model with multiple axonal compartments to maintain appropriate timing of communication among model elements. Five to twenty axonal compartments were incorporated into the pelvic nerve projections so that afferent information from the bladder arrived at the rostral receptive sites in a realistic sequence reflecting physiological transmission times that communicate peripheral afferent information from the bladder and sphincter to the PSC and PAG. The afferent and efferent information transmitted through the pelvic nerve was shown as distinct pathways even though the information travels in a single nerve. The properties of each neuron and the weighting variables associated with the network are described in the S1 Materials. When a node in the traditional model (Fig 1A) simply passed information without internodal interactions, we did not model the neuronal dynamics of these, unnumbered nodes; we passed the information directly to the next node and added a time constant to reflect the processing time of the unnumbered nodes in Fig 1.

## 2.5 Modeling the effect of TMS pulses

Transcutaneous magnetic pulses applied to the lumbar spinal cord were assumed to affect the sodium conductance within targeted neurons as the means of activating specific neurons in the model (Nodes 1, 2, and 3; Fig 1B), where sodium channel activity was modeled as follows:

$$I_{Na} = (g_{Na} + g_{TMS})(v - v_{Na}) \tag{2}$$

In this formulation, the effect of TMS is represented as an additional sodium channel opening through the conductance variable $g_{TMS}$. The governing equation that we used was

$$g_{TMS} + \tau\dot{g}_{TMS} = ku \tag{3}$$

where $u$ and $g_{TMS}$ are time-dependent variables representing the TMS pulse, and the portion of sodium channel conductance affected by the TMS stimulation, respectively; $\dot{g}_{TMS}$ is the rate of change of $g_{TMS}$, $k$ is the gain (roughly proportional to the inverse square of the distance between the source of TMS (a figure-8 coil over the lumbar spine) and the neuron, and $\tau$ is the time constant for closing the sodium channel affected by each TMS pulse. Activation of sodium channels as a mechanism of action of TMS is consistent with findings in acute

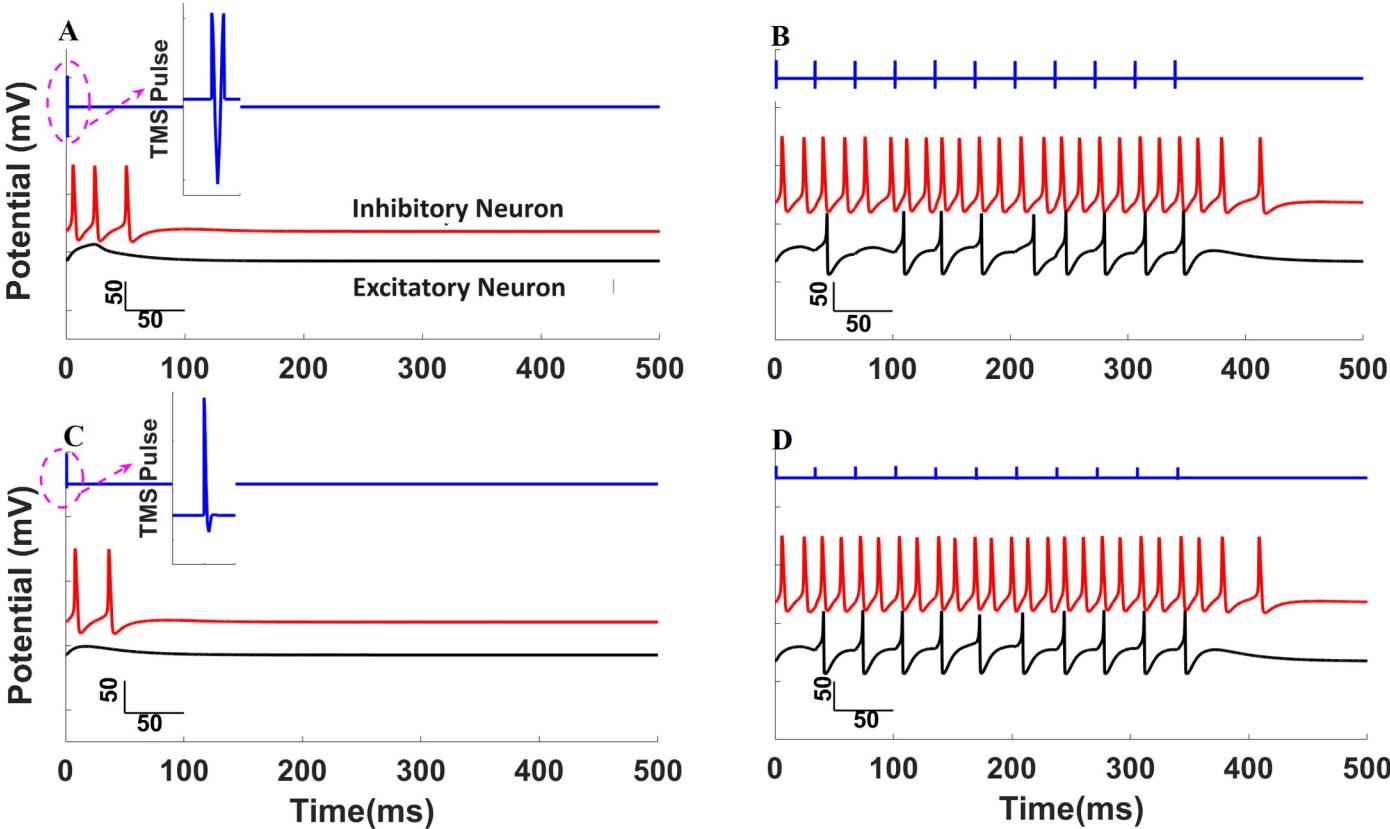

**Fig 2. The effect of low and high frequency, biphasic and monophasic TMS on inhibitory and excitatory interneurons.** In each plot, the electromotive forces associated with biphasic or monophasic TMS pulses are plotted (blue lines, first trac: biphasic A and B; monophasic C and D). The inhibitory interneuronal responses in the lumber spine (Node 2) are plotted in red (second trace), and excitatory lumbar neuronal responses (Nodes 1 and 3) are shown in black (third trace). The responses to 1 Hz TMS are shown on the left (top panel biphasic stimulation and bottom panel monophasic TMS), and the neuronal responses to 30 Hz TMS are shown in the right panels. For the same amplitude TMS, biphasic stimulation was more effective (elicited more action potentials) than the equipotent monophasic stimulation. Inhibitory neurons were more sensitive to the TMS stimulation than excitatory neurons so that for any level of TMS, more action potentials were elicited from the inhibitory neuron. High frequency pulses had cumulative effects (see text) and created bursts of activity, which were both at a higher frequency and lasted slightly longer in the inhibitory neurons. These differences in sensitivity to TMS led to important differences in response to 1 Hz or 30 Hz TMS within the enhanced model of control of micturition. This figure was created by MATLAB.

experiments in which TMS pulses applied to rat cortical brain slices caused immediate activation of voltage-gated sodium channels and induced current flow into the neuronal soma of cortical neurons [14]. Nodes 1 and 3 represent excitatory interneurons (green) and Node 2 is an inhibitory interneuron (red). During lumbar spinal TMS, a monophasic or biphasic TMS pulse train was applied to Nodes 1, 2, and 3, as shown in Fig 1B, as the input signal $u$ in the model equation. The effect of a TMS pulse on each type of node (inhibitory or excitatory) was simulated using the single-compartment model with $\tau$ = 30 ms, which is relatively long, to elicit bursting activity after TMS (the logic of this selection is discussed below).

## 2.6 Effects of TMS at 1 or 30 Hz on excitatory/inhibitory nodes 1, 2, and 3 within the lumbar spine

Before assessing the impact of lumbar TMS on the circuit controlling micturition, we analyzed the responses of the individual, recipient, interneurons in the lumbar spine during 1 Hz and 30 Hz TMS. The neural activities when trains of TMS pulses were applied are summarized in Fig 2. We examined various types of stimulation with different patterns of the electromotive

force (EMF) derived from TMS (biphasic—panels A and B in Fig 2 and monophasic—panels C and D in Fig 2) and different TMS frequencies (1 Hz—left panels of Fig 2 and 30 Hz—right panels of Fig 2). The coil current (and the induced magnetic flux) of the TMS pulse is biphasic, but the EMF induced by the flux is "tri-phasic" since the EMF is the derivative of the applied stimulation. Therefore, the EMF associated with either biphasic or monophasic stimulation is shown in blue in panels A and C. The TMS pulses were set so that the peak value of $ku$ was roughly 4 $k\Omega^{-1}/cm^2$ (where $u$ is a time-dependent variable representing the effect of the TMS pulse in terms of EMF; $k$ is the gain), and the pulse width was 210 μs for both biphasic and monophasic stimulation pulses. These values were used for all TMS simulations that we conducted.

Low frequency TMS (1 Hz) generated short duration bursts of activity in the inhibitory node that decayed back to the steady-state level before the next pulse was applied Fig 2A and 2C, second, red tracing). In contrast, low frequency TMS failed to generate action potentials at the excitatory nodes (Fig 2A and 2C, third, black tracing) since the intensity of the TMS pulse did not reach the threshold for action potential generation [41]. These responses were similar for both biphasic and monophasic TMS pulses (Fig 2A and 2C, respectively). Thus, inhibitory neurons were more excitable and sensitive to low frequency TMS pulses than excitatory neurons; the difference in excitability originated mainly from differences in the potassium channel conductance—a smaller value of $g_K$ in the inhibitory node [41–43].

High-frequency TMS (30 Hz) demonstrated a similar pattern of responses in terms of sensitivity of the inhibitory/excitatory properties of the nodes, as seen on panels B and D in Fig 2. The inhibitory node generated a train of action potentials (second, red tracings) with a spike frequency greater than the spike frequency of the excitatory node after both biphasic and monophasic TMS (third, black tracings). When the train of TMS pulses ceased, the action potentials were sustained longer in the inhibitory node than in the excitatory node(s). However, the cumulative effect of multiple high-frequency TMS pulses was quite different from the response to low-frequency TMS due to the long time-constant used in modeling the neuronal effect of TMS on the sodium conductance: the effect of several TMS pulses accumulated during higher-frequency stimulation so that the response of the excitatory node was entrained to the TMS pulses. The inhibitory node generated a spike train at a frequency higher than the frequency of the applied TMS because the time constant of the neuronal response to TMS in the inhibitory node was sustained longer than the interval between the two positive spikes of electromotive force within a single biphasic TMS pulse, and the effects of these serial spikes within a single biphasic pulse of TMS summed. In contrast, 30 Hz TMS was not sufficient to exceed the threshold to generate a train of spikes in the excitatory neuron, and the excitatory neuron simply generated one spike per stimulation. Because the biphasic pulse was more effective generating action potentials in inhibitory nodes at 1 Hz stimulation, we used biphasic stimulations for all subsequent simulation studies of the neuronal circuits.

## 3. Results

Within the enhanced model, we tested three hypotheses: first, we tested the hypothesis that the expanded model could replicate normal bladder behaviors, and second, we tested the hypothesis that responses to SCI could be reproduced by the model. Third, we tested the hypothesis that TMS below the level of SCI may ameliorate different aspects of voluntary control of micturition with the same goal as de Groat and Wickens [13]–using modeling to validate the underlying neurophysiology of urine storage and voiding after SCI and during TMS in patients with SCI.

### 3.1 Healthy storage/voiding functions

The first step in validating the enhanced model was to confirm that the circuitry of the enhanced model still replicated normal patterns of urine storage and micturition—to verify that the additions to the model did not disrupt or preclude normal control of micturition. Neural activity patterns within selected nodes in the enhanced model controlling micturition in a healthy subject during the storage phase are shown in Fig 3. A schematic of the circuit elements active during storage (highlighted—thicker lines) is shown in Fig 3A. A similar schematic of the circuit elements active during voiding (highlighted) is shown in Fig 3B. Examining storage first, the efferent activity of the hypogastric nerve at Node 4 is shown in Fig 3C1. The activity of the hypogastric nerve is a function of bladder filling, and therefore, the activity in Node 4 also reflects the afferent activity of the pelvic nerve that would also be recorded from Node 5. During the simulation, the bladder filled faster than normal to make the effect of bladder filling on hypogastric nerve activity visible in a short time. Hence, the frequency of efferent hypogastric nerve action potentials increased from 10 to 45 Hz sequentially in three large steps (Fig 3C1; steps in bladder volume indicated by arrow heads below the tracing of nodal activity). As the bladder filled and as afferent signaling from the bladder increased, parasympathetic activity that might contract the detrusor muscle was inhibited (Node 8, Fig 3C2) [44,45], and efferent activity from motor neurons in the Nucleus of Onuf, fostering external sphincter contraction, increased as a function of bladder filling (Node 9, Fig 3C3). Hence, urine was retained appropriately in the relaxed bladder.

When the firing frequency of the afferent fibers in the pelvic nerves increases as the bladder fills and reaches a threshold, micturition is initiated, which activates the PAG in the absence of inhibitory commands from higher centers (HC; Fig 3B) not to urinate. During micturition, sympathetic detrusor inhibition was inactivated; Node 4 was quiet in Fig 3D1; sphincter excitation was withdrawn (Fig 3D3; Node 9); and detrusor contraction was activated (Fig 3D2; Node 8). As a result, coordinated contraction of the bladder and relaxation of the sphincter led to and permitted bladder emptying. Thus, the enhanced model simulations captured the storage/voiding mechanisms observed in healthy subjects despite the addition of novel circuit elements. It is worth noting that the inhibition of the latent spinal voiding reflex by Node 6, active in normal, uninjured individuals, was essential to prevent the emergence of infantile, reflex (volume-controlled) micturition.

### 3.2 Condition after SCI above T10: Detrusor-sphincter dyssynergia

An upper-level SCI blocks communication between the PMC/PSC and the neural circuits in the lumbosacral spinal cord below T10 (red dotted line), and without descending signals from the pontine nuclei, the LUT was controlled solely by local spinal reflex circuits. Simulation of this local spinal control is represented in Fig 4B1-B3 when the bladder was almost full. The pelvic nerves sent a train of afferent action potentials at 40 Hz, which increased efferent activity in Node 4 (Fig 4B1). However, the afferent feedback could not reach the PAG or the PMC due to the SCI, and the descending commands to Nodes 1, 2, and 3 were inactive (or absent). Consequently, Nodes 4, 8, and 9 were activated simultaneously by local spinal reflex mechanisms, including the infantile spinal voiding reflex, as indicated by the spiking activity in Fig 4B1-B3. Simultaneous inputs to the detrusor muscle from both inhibitory sympathetic (Node 4, Fig 4B1) and excitatory parasympathetic neurons (Node 8, Fig 4B2), respectively, may result in sporadic contraction/relaxation as has been observed in patients following SCI [6,13]. Moreover, bladder afferent activity simultaneously activated the external sphincter (Node 9, Fig 4B3) since reciprocal supraspinal regulation of detrusor and sphincter muscle activities was no longer possible after SCI. Thus, the model successfully reproduced the uncoordinated,

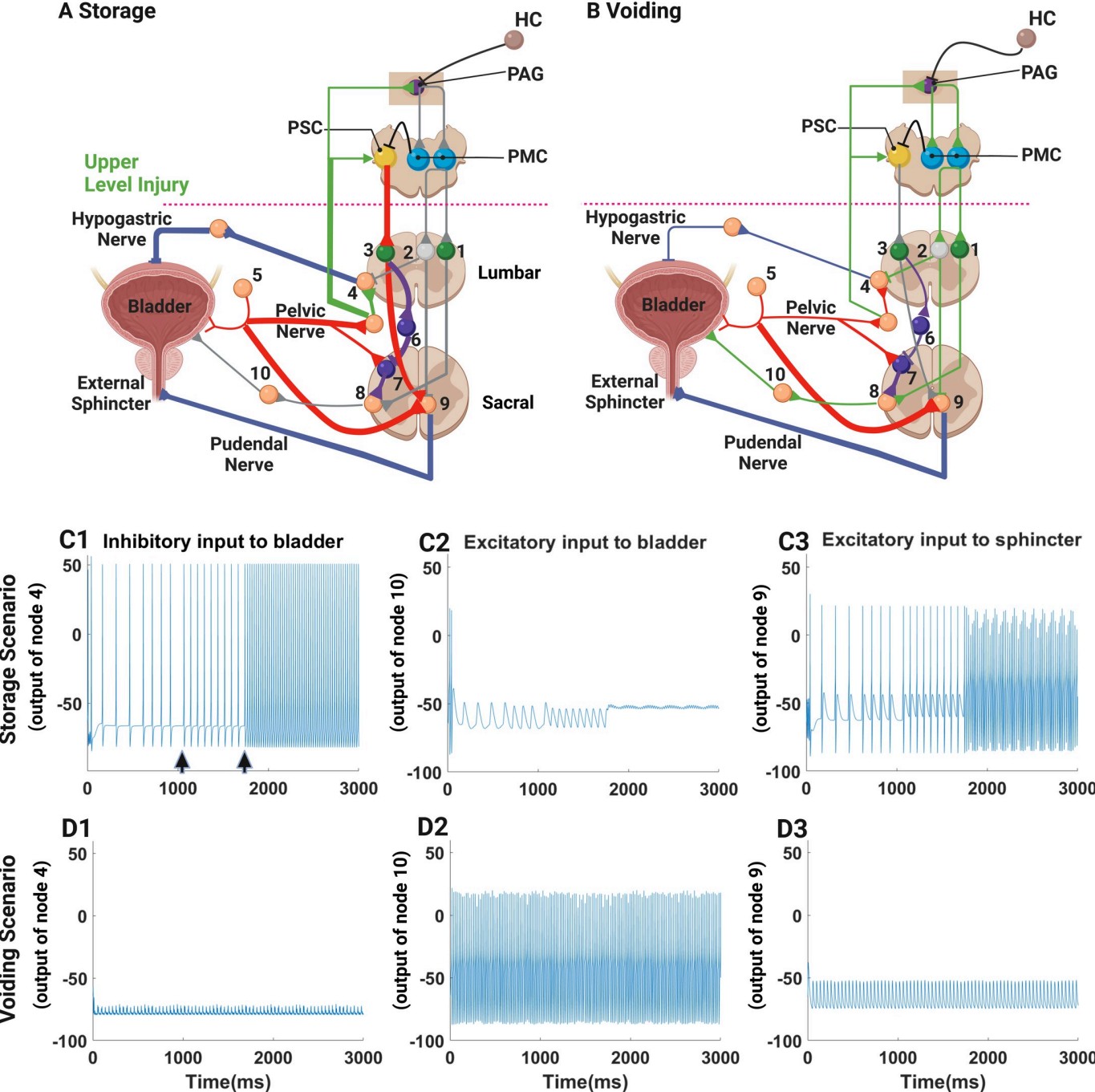

**Fig 3.** Enhanced model simulation of normal bladder control during storage (A) and voiding (B). The elements of the network controlling storage and voiding are highlighted (thick lines) in A (storage) and B (voiding). The activities of Nodes 4, 8 and 9 are shown sequentially in separate panels (1–3). The activity shown represents the simulated changes in membrane potential at each node, which resembles somatic neuronal activity. The activities of Nodes 4, 8 and 9 during urine storage (A and C1-3), Nodes 4 simulates volume-dependent sympathetic inhibition of detrusor contraction (C1; note that the bladder volume was increased in three large steps—each step indicated by an arrowhead below the tracing of nodal activity and each step associated with a commensurate increased in nodal activity); parasympathetic activity stimulating detrusor contraction, which is inhibited during urine storage is shown in C2; and sphincter contraction originating from the Nucleus of Onuf as represented by Node 9 as shown in C3. Activity of the PSC and inactivity of the PAG during urine storage mean that detrusor muscle relaxes and sphincter muscle contracts. During voiding, the PAG is activated, which in turn activates the PMC, the detrusor contracts (note high activity in the parasympathetic preganglionic Node 8 (D2), and the sphincter relaxes so that activity in Node 9 is reduced. In order to permit the detrusor to contract, sympathetic inhibition of the detrusor is withdrawn, and Node 4 is quiet, thereby enabling urine to flow out of the bladder during detrusor contraction. This figure was created in part with BioRender.com.

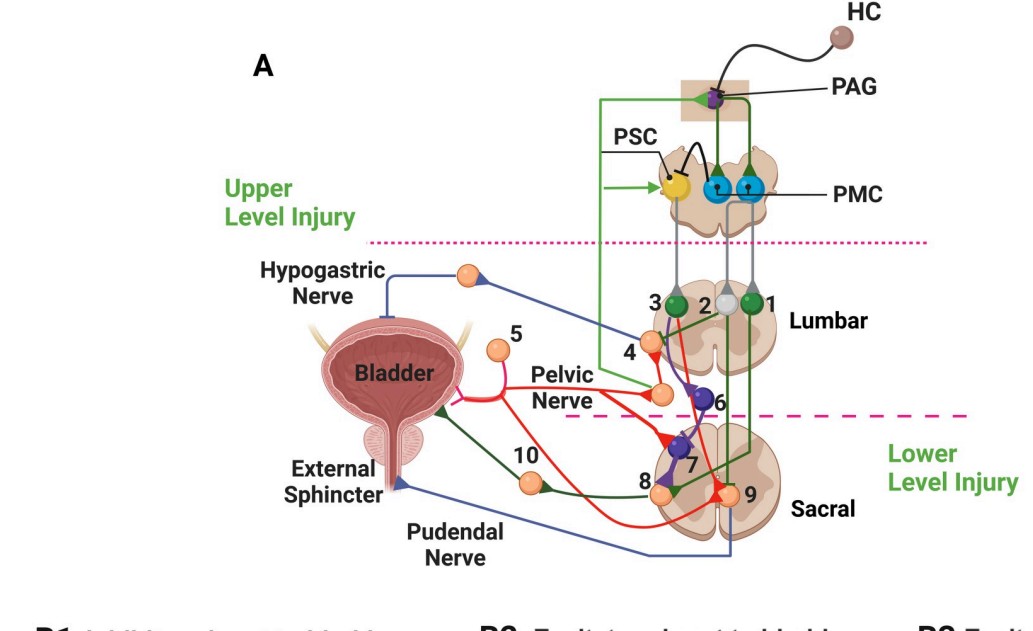

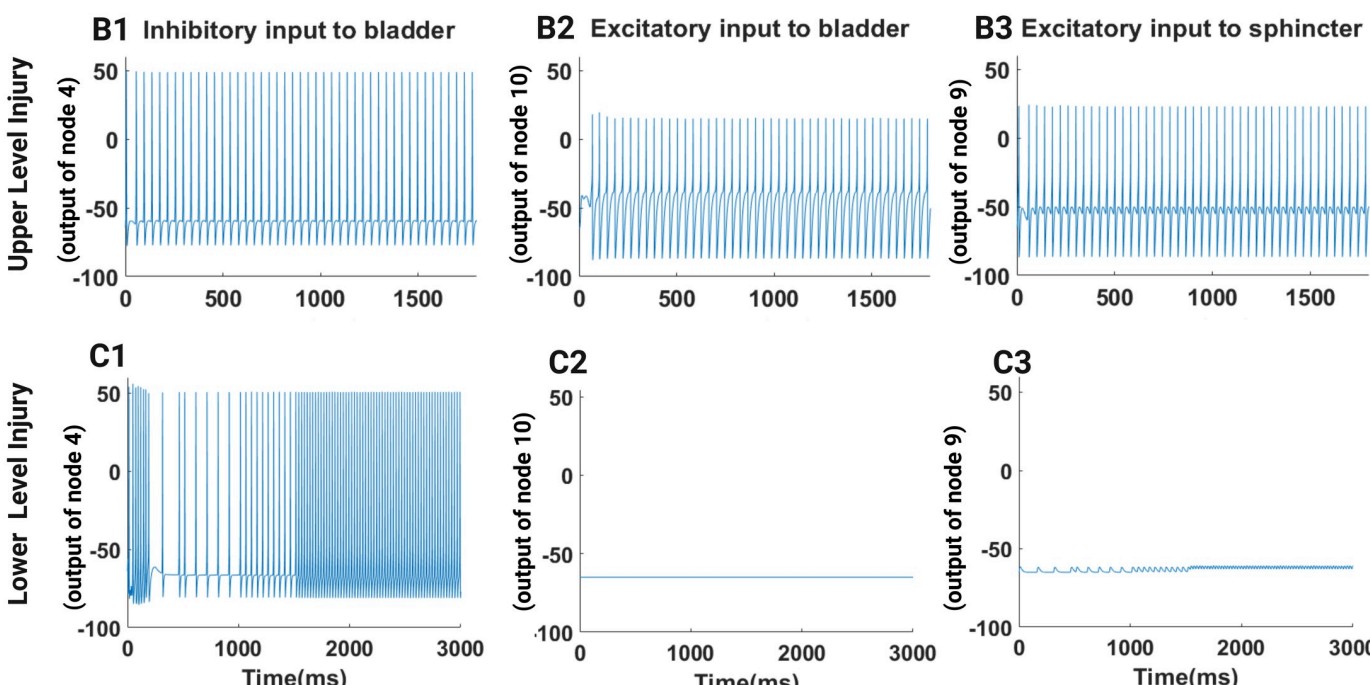

**Fig 4.** Simulation based on enhanced model replicates detrusor-sphincter dyssynergia after upper-level SCI (B) and hyperactive bladder activity after lower-level SCI (C). The schematic of the enhanced neural circuit of micturition with both an upper-level SCI (green dashed line) and a lower-level SCI (red dashed line) shown. B: After an upper-level SCI, supraspinal control of micturition is lost, and both voiding and storage reflexes are active, causing the detrusor-sphincter dyssynergia. The activity of Nodes 4, 8 and 9 are shown in B1. 2 and 3, respectively. Note that as the bladder is full, afferent activity from the bladder drives sympathetic (Node 4) to keep the detrusor muscle relaxed. However, there is no input from the PSC to inhibit activity of the preganglionic parasympathetic neurons (Node 8) and the detrusor receives active parasympathetic drive to contract the muscle. Simultaneously, the motor neurons driving sphincter contraction, represented by Node 9, are active. As a consequence, the bladder and external sphincter contractions occur simultaneously and the coordination of bladder sphincter activity necessary to empty the bladder is lost. C: After a lower-level SCI, sympathetic activity increases (Node 4, C1) as the bladder is filled (the time scale of filling has been accelerated), but there is no activation of detrusor contraction (Node 8, C2), and no external sphincter contraction (Node 9). As a result, the bladder leaks continuously as it is filled. This figure was created in part with BioRender.com.

simultaneous activation typical of detrusor-sphincter dyssynergia and overactivity observed after SCI that prevents voiding and disrupts voluntary control of micturition when the bladder is wholly or partially full [46].

## 3.3 Condition after lower-level SCI: Urinary incontinence

When a spinal cord injury occurs at the conus medullaris, the sacral region of the neural circuit controlling detrusor and external sphincter contraction can be severely damaged. Such a low-level SCI prevents excitation of the detrusor and the sphincter muscles (Fig 4A, red dashed line) and leads to low bladder pressure, low external sphincter tone, and constant dripping of urine [47,48]. The results of simulating a lower spinal cord injury are shown in Fig 4C1-C3. To simulate a lower spinal cord injury, we made Nodes 8 and 9 inactive and unresponsive to synaptic inputs. Bladder filling was simulated in the model, and afferent sensory signals from the bladder increased as the bladder was filled. Inhibitory activity originating from Node 4 (Fig 4C1) suppressed detrusor contraction, and there was no activation of the detrusor (Node 8 was inactive, Fig 4C2) or external sphincter (Node 9 was quiet as well; Fig 4C3). In the presence of ongoing bladder filling, continuous inhibition of the detrusor muscle without contraction of the sphincter muscle permitted constant leakage of urine from the flaccid bladder. Thus, the model replicated the dripping incontinence observed after a lower-level SCI.

## 3.4 Acute effect of TMS on bladder function after SCI above T10

We examined the effects of TMS at low (1 Hz) and high (30 Hz) frequencies in the presence of a SCI above T10. A train of 1 Hz TMS pulses was applied to Nodes 1, 2, and 3 within the lumber spine, and the resulting neural activities in Nodes 4, 8, and 9 were simulated, as shown in Fig 5,C1-C3. The inhibitory, lumbar interneuronal node was more easily activated by 1 Hz TMS pulses, while no action potentials were generated in excitatory, lumbar Nodes 1 and 3. During 1 Hz stimulation, the activity of Node 2, acting through spinal interneurons, caused episodic inhibition of sphincter contraction (Fig 5C3, Node 9) and detrusor muscle relaxation (Fig 5C1, Node 4). Since excitatory Nodes 1 and 3 were inactive during 1 Hz stimulation, the infantile, spinal voiding reflex was activated in the absence of descending inhibition from the PSC acting through Nodes 3 and 6, which permitted detrusor activation based on afferent information directed to Node 7 and then efferent activation of detrusor contraction through parasympathetic preganglionic neurons (Node 10, Fig 5C2). This sequence of coordinated external sphincter relaxation and persistent detrusor activation occurred repeatedly a few hundred milliseconds after each 1 Hz TMS pulse was applied at time t = 0 and 1000 ms for slightly less than a 100 ms, as shown in Fig 5C1-C3). Thus, the acute effect of 1 Hz TMS alleviated detrusor-sphincter dyssynergia and facilitated intermittent voiding.

During high frequency stimulation at 30 Hz (Fig 5D1-D3), excitatory and inhibitory nodes (Nodes 1 and 3 and Node 2, respectively) were activated. Excitation of Node 2 inhibited sympathetic activity in Node 4, as shown in Fig 5D1. The excitatory signals from Nodes 1 and 3 had antagonistic effects on the spinal voiding reflex: Node 3 inhibited, and activity in Node 1 promoted activation of the detrusor muscle. Hence, bursting activity in Node 10 (Fig 5D2), representing preganglionic parasympathetic activity to contract the detrusor muscle, was sporadic and elicited irregular spiking and intermittent detrusor contraction (Node 10, Fig 5D2). Moreover, excitation of Node 3 activated inhibitory Node 6 and prevented activation of the spinal voiding reflex that might otherwise have coordinated detrusor contraction and external sphincter relaxation and effective voiding. However, 30 Hz TMS evoked high-frequency spiking in Node 9 (Fig 5D3), and constant external sphincter contraction, thereby obstructing urine flow and preventing micturition and favoring continence, which mimicked the motor

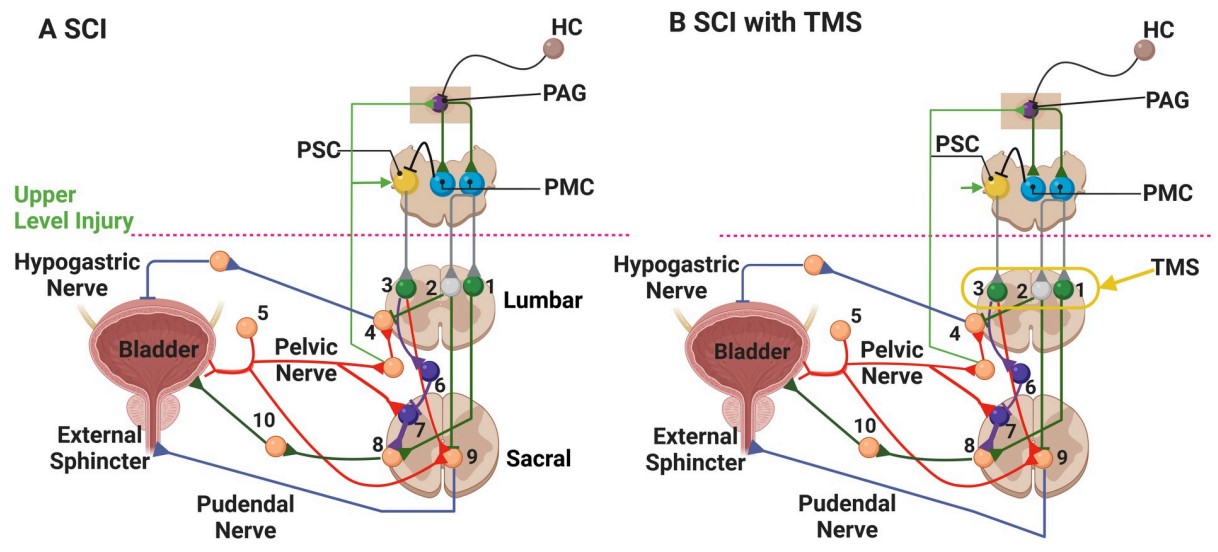

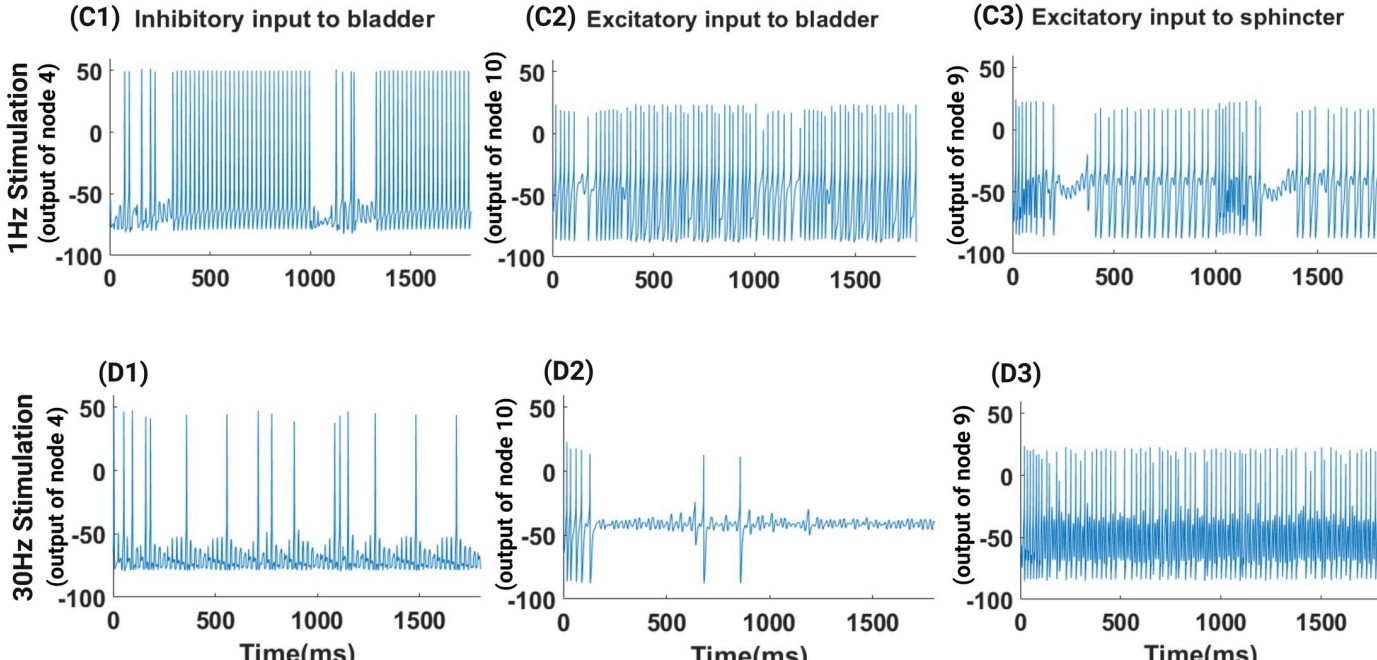

**Fig 5. Modeling the effect of TMS in the enhanced model of control of micturition.** The response to 1 Hz TMS is shown in C1-3. The timing of TMS is shown by arrows, and after each arrow, the bladder briefly contracts, and sympathetic inhibition of detrusor contraction ceases briefly (Node 4, C1), and the external sphincter is relaxed briefly (Node 9, C3). As a consequence, the parasympathetic drive to the detrusor contracts the bladder (Node 10, C2). In contrast, 30 Hz TMS suppresses sympathetic drive to the bladder (Node 2 inhibits Node 4, D1); excitation of Node 1 inhibits preganglionic parasympathetic activity, and the detrusor lacks activation to contract (Node 10, D2); while excitatory Node 3 drives the Nucleus of Onuf and Node 9 activity (D3) results in contraction of the external sphincter. This figure was created in part with BioRender.com.

activity typically associated with urine storage in the bladder. Thus, the differential sensitivity of the lumbar, inhibitory, and excitatory interneuronal nodes (Node 2 versus Nodes 1 and 3, respectively) to 1 Hz and 30 Hz TMS led to distinctly different bladder/voiding behaviors within the model during low and high-frequency TMS after SCI at T10 or above.

## 4. Discussion

We developed a computer model describing control of micturition to determine what attributes of the neural system controlling micturition were necessary to replicate both normal and pathological bladder behaviors after SCI so that the responses to lumbar TMS might be understood in patients after SCI. To achieve this goal, we started with a traditional model that described control of micturition in normal subjects. Next, we enhanced the traditional model by adding novel features that were necessary to replicate pathological bladder behaviors after SCI and necessary to replicate the changes in bladder function elicited acutely by TMS in patients after SCI. de Groat et al. [49] proposed four likely processes involved in neuroplasticity after SCI: 1. A low thoracic SCI eliminates bulbo-spinal inhibitory reflexes; 2. The strength of remaining synapses might be altered by new synapse formation; 3. Neurotransmitter concentrations might be changed, and 4. Alterations in afferent inputs might develop. These suggested mechanisms are close to the essential enhancements that we found were necessary to reproduce pathological bladder responses after SCI. We simulated the loss of bulbo-spinal inhibition of spinal reflexes and increased the complexity of the lumbar spinal interneuronal network (this is not strictly in the list above because it involved creating the circuitry of the latent spinal voiding reflex) and introduced an element of neuroplasticity after SCI, reflected in increased synaptic strengths within the spinal voiding reflex (representing either new synapses or altered neurotransmitter release—these processes are lumped into synaptic strength in our model) to allow more nuanced processing of sensory afferent information at the spinal level; the afferent inputs to the spine were unchanged (contrary to item 4. above). These additions to the traditional model permitted more varied efferent control of bladder function and permitted the enhanced model to replicate both normal and pathological bladder function.

Before discussing the findings in the paper, it is worth emphasizing the effect of two assumptions inherent in the traditional and enhanced models. First, there are few to no single cell recordings (current clamp or patch clamp) of spinal interneurons that have identified unique, functionally specific, ionic currents in the neurons that support micturition. Therefore, we used typical conductances in the Hodgkin-Huxley formulations for each node, and these generic neurons behaved as logic gates passing excitatory or inhibitory activity to the next node of the network, much as others have done when modeling control of micturition [12]. Second, we assumed that all sources of afferent information were equal and similar regardless of receptors generating the information and the axonal velocities of nerves conducting the information. These two assumptions mean that all the enhancements necessary to replicate bladder dysfunction after SCI and the response to lumbar TMS are embodied only in changes in interneuronal populations (adding generic interspinal neurons), increasing the complexity of interneuronal network connections, and changes in synaptic strength. That non-specific neurons at the nodes of the model can simulated the control of micturition indicates that the patterns of neuronal connectivity and the nature and strength of the synaptic connections (inhibitory or excitatory) are likely the dominant elements of bladder control, which is consistent with the previous experimental studies. The models proposed are no more complex than they need to be to reproduce the observed behaviors after SCI and after TMS in the setting of SCI, but the models are also not comprehensive descriptions of bladder control. As mathematical models seek to represent more nuanced interactions between multiple sources of afferent and efferent control of bladder function, the elements of those models will have to incorporate more physiologically realistic and complete representation of the receptors, neurons and synaptic diversity involved in controlling micturition.

Two key findings emerged from the enhanced model with respect to control of micturition after SCI. First, the results of model simulations confirmed the critical role of an infantile

spinal voiding reflex. The enhanced model reproduced detrusor-sphincter dyssynergia, a typical pattern of bladder dysfunction after a SCI above the lumber level, by blocking or weakening inhibitory signals from the PSC and by strengthening the spinal voiding reflex, which likely emerges or re-emerges—since the reflex is present in infants—as an effect of post-SCI neuroplasticity [50], which we modeled by changing the synaptic weights among the interneurons regulating the spinal voiding reflex. Second, differential sensitivity of excitatory and inhibitory lumbar interneurons during TMS was a necessary feature of the model [41,51–53], which allowed low-frequency stimulation to inhibit reflex regulation of external sphincter activity selectively so that activation of the spinal voiding reflex could elicit detrusor contraction coordinated with the relaxation of the external sphincter to permit bladder emptying. The differential sensitivity of inhibitory Node 2 to low-frequency stimulation meant that high-frequency stimulation, which activated both inhibitory and excitatory nodes (Node2 and Nodes 1 and 3) in the lumbar spine, did not enhance micturition because detrusor contraction was inhibited, and sphincter contraction was excited by TMS-mediated lumbar activation of interneuronal circuits, thereby favoring storage of urine.

## 4.1 Mechanisms underlying the detrusor-sphincter dyssynergia after SCI

Detrusor-sphincter dyssynergia and overactive bladder emerge from the activity of autonomous spinal mechanisms that control detrusor and sphincter functions without descending inputs from the PMC, and PSC, which cannot participate in the control of micturition after a SCI. A spinal mechanism for autonomous bladder contraction is thought to re-emerge after SCI through neural reorganization [6]. Micturition in infants happens involuntarily through the activation of a spinal voiding reflex that is initiated when the bladder fills to a threshold value [35]. As infants mature, neural circuits in the pons and cortex develop, inhibit the spinal voiding reflex, and regulate the voiding reflex through actions of the PAG to allow volitional voiding [6,13]. Moreover, a spinal voiding reflex was activated by C-fiber bladder afferents in chronic spinal cats several weeks after transection [32]. Volume-related bladder stretch information is usually transmitted to the spine by myelinated Aδ fibers, and C-fibers, though they are present in pelvic nerve afferents, are normally silent in the absence of inflammation, irritation, or overdistention [54]. Following a SCI, however, the C-fibers emerge as a more potent afferent mechanism triggering intraspinal reflex bladder contractions [1]. Much as a spinal voiding reflex emerged in cats after spinal cord transection, we incorporated a re-emergent infantile spinal voiding reflex within the lumbosacral spinal cord that is latent and inactive or undetectable due to inhibition by the PSC in normal human adults (though we did not represent any C-fiber activity explicitly in the model). The enhanced model shown in Fig 1B, incorporates descending inhibition from the PSC to explain normal urine storage and the activity of PAG and PMC to control voiding in healthy adults, but adds to the model features necessary to explain the observed pathological behaviors associated with micturition after SCI. We assumed that neural plasticity strengthened the spinal voiding reflex, and since the descending inhibitory signals to the lower spine from the PSC and PMC were absent after SCI, the infantile spinal voiding reflex was reactivated while the spinal sphincter storage reflex was maintained. In this setting, both detrusor and sphincter muscles can contract when afferent sensory feedback passes a threshold reflecting sufficient bladder filling to justify voiding. Thus, equipped with the addition of a spinal voiding reflex, the proposed model reproduced detrusor-sphincter dyssynergia as it is typically observed in patients after SCI.

## 4.2 Modeling the effect of TMS as sodium channel activation

Modeling efforts to simulate the neuronal effects of TMS have focused on the electromagnetic field imposed on the neural tissue and not on the effect of TMS on neural activities [55–57]. However, patch-clamp experiments indicated that TMS activated voltage-gated sodium channels [14,52], and this result provides a plausible way to model the impact of TMS on the spine as a transient change in the conductance of ion channels [51,53], which we incorporated within each neuronal node in the spinal region affected by TMS (Nodes 1, 2 and 3).

Because we simulated the behavior of assemblies of neurons with single nodes, it is difficult to capture the time dependencies of responses to TMS in networks of neurons. A TMS pulse induced a transient inward current in neuronal cells that decayed with a time constant of 3.08 ms [14,53,58]. As a result, a single action potential was generated immediately after each TMS pulse, which is consistent with experimental measurements from patch electrodes [52]. During extracellular recordings of parietal cell activity in monkeys, single TMS pulses elicited a short latency (10 ms after the TMS pulse) burst of action potentials in individual neurons, and the bursts lasted less than 50 ms [59]. The bursts were restricted to relatively small numbers of neurons within the region of TMS. However, there were other neurons in which a TMS pulse produced an excitation-inhibition-excitation (or burst-silence-burst) pattern lasting about 250 ms. The bursting after a TMS pulse appeared to be an immediate effect of TMS propagating through the local neuronal network [59]. Based on the patch-clamp study [14], the time constant, $\tau$, of closing of sodium channels after a TMS pulse should be a few milliseconds, but $\tau = 30$ ms was used in our model to capture the network effect within each node. We attribute the large time constant to network effects, but it may also result from additional effects of TMS on other ion channels within the node. The presence of a long-time constant in the node generated a train of action potentials in the inhibitory node in response to a single TMS pulse (Node 2), as indicated in (Fig 2). Thus, the long-time constant is a mechanism of achieving prolonged bursting in the model, which is based on the behavior of individual nodes in the model, when in reality, this behavior is likely achieved in actual neuronal tissue by network interactions among populations of neurons and these inter-neuronal interactions likely involve calcium and potassium channel dynamics and other biophysical properties of neurons and networks (rather than a change in a single, nodal time constant). We imposed these dynamic properties on the node within the area affected by the magnetic field generated by TMS to achieve a bursting response to pulses of TMS.

## 4.3 Selective activation of neural circuits by TMS

Within the model, excitatory and inhibitory neurons responded to TMS differently: the inhibitory node(s) were more sensitive to TMS than the excitatory nodes. The inhibitory node, Node 2, responded to a single TMS pulse with a train of action potentials (Fig 2), but the excitatory nodes was unresponsive to low-frequency stimulation (Nodes 1 and 3). This property of our model is based on the finding that some cortical inhibitory interneurons have a low threshold for spiking [60,61], and the intrinsic excitability of inhibitory neurons is greater than that of excitatory neurons [53,62]. Based on these observations, Pashut et al. [51,52] hypothesized that low-frequency TMS selectively activated inhibitory neurons. More recently, Mahmud and Vassanelli [41] investigated the differential modulation of excitatory and inhibitory neurons following extracellular electrical stimulation and emulated the effect of stimulation using sinusoidal current injection into model neurons, which were based on Hodgkin-Huxley equations. Model parameters were selected to emulate the dynamics of regular-spiking, excitatory pyramidal neurons [42] and fast-spiking inhibitory interneurons [43]. Stimuli with specific frequencies and amplitudes differentially enhanced or suppressed excitatory or inhibitory

neurons. Selective activation by TMS pulses applied at different frequencies of inhibitory or excitatory nodes in the model allowed us to treat different dysfunctional aspects of micturition observed after SCI. The different excitability in our model of inhibitory and excitatory nodes was mainly caused by the magnitude of the potassium channel conductance (the inhibitory neurons had a lower potassium conductance). A general conclusion, also supported by passive cable modeling of TMS effects [51], is that neurons with a low threshold for membrane excitability after current injection are more easily activated by electromagnetic stimulation.

Many forms of stimulation to treat detrusor-sphincter dyssynergia have failed because they activate only a single element of the complex three-pronged effects of the spinal micturition circuit (parasympathetic stimulation of the bladder, withdrawal of sympathetic stimulation of the detrusor, and inhibition of somatic motor neuronal activation of the external sphincter). In contrast, the different sensitivity of inhibitory and excitatory neurons to TMS can be exploited to mitigate detrusor-sphincter dyssynergia after SCI by modifying all three elements of effective reciprocal control of micturition. Inhibitory neurons are sensitive to both low- and high-frequency TMS, but excitatory neurons are only sensitive to high-frequency TMS in the proposed model (Fig 2). When low frequency (1 Hz) TMS pulses were applied at the L1 level of the spine (Fig 5C1-3), inhibitory Node 2 was activated and prevented bladder relaxation while simultaneously inhibiting external sphincter contraction, but excitatory Nodes 1 and 3 were inactive, thereby disinhibiting the spinal voiding reflex. Thus, the acute effect of 1 Hz stimulation was relaxation of the sphincter muscle and contraction of the detrusor muscle, which resulted in voiding. On the other hand, 30 Hz stimulation activated both inhibitory Node 2 and excitatory Nodes 1 and 3, which resulted in sphincter contraction and detrusor relaxation and did not promote or permit voiding. The role of Node 6 (Fig 1B), which represents a population of inhibitory interneurons, is important since low-frequency TMS cannot effectively elicit voiding if the spinal voiding reflex is directly inhibited by TMS-mediated activation of Node 3 (or an excitatory input from the PSC to Node 3 in intact, normal subjects), which activates inhibitory Node 6. The outcomes of the model-based study for both 1 Hz and 30 Hz stimulations were consistent with the experimental results in human subjects with SCI treated with serial TMS [2]. The differential sensitivity of excitatory and inhibitory interneurons to TMS in the model may explain why low-frequency TMS treatment more effectively restored voluntary control of micturition than 30 Hz TMS in patients after SCI [2]. Finally, the model output is consistent with the principle that the differential sensitivity of inhibitory and excitatory neurons is crucial for the selective activation of bladder contraction and sphincter inhibition via pudendal nerve stimulation [12,63–65].

Transcutaneous electrical stimulation also improved incontinence and dyssynergia in patients with SCI [3,66]. Serial treatment twice a week with transcutaneous 30 Hz electrical stimulation between T11 and L1 for 8 weeks increased bladder capacity and reduced symptoms of incontinence and dyssynergia—an effect that was attributed, like TMS, to simultaneous effects on both the afferent and efferent limbs of sacral control of micturition [66]. Transcutaneous electrical stimulation at 30 Hz reduced detrusor-sphincter dyssynergia, which is at odds with the simulations presented (Fig 5D1-3) and with the findings presented by Niu et al. [2]. In addition, unlike TMS, an 8-week course of therapy using transcutaneous electrical stimulation did not improve bladder efficiency or restore volitional control of micturition, and though there is a claim that chronic electrical stimulation improved 'engagement' with more rostral, supraspinal elements controlling micturition, there were no measurements to indicate how engagement might be measured as an index of improved bladder control [66]. The reasons for these differences will require additional animal studies and perhaps further refinements to the enhanced model of bladder control.

## 4.4 Broader implications and limitations

There is little information about the specific biophysical properties of spinal neurons governing control of voiding. Therefore, it is useful and instructive to construct a mathematical framework within which we can test hypotheses that may enhance our understanding of the potential mechanisms underlying bladder dysfunction after SCIs, such as detrusor-sphincter dyssynergia. The model creates an environment in which we can propose the minimum necessary neuronal substrates required to replicate observed clinical problems and assess different therapeutic patterns of spinal cord stimulation and their effects on micturition. Modulating spinal cord circuits by electrical or magnetic stimulation can ameliorate a variety of dysfunctional conditions after SCI, such as hand motor control [67], locomotion [5], neurogenic bladder [2], sensorimotor system [68], neuropathic pain [69], and spasticity [70]. However, the mechanisms underlying the immediate and long-term effects of stimulation on neural activity at the single-cell and network levels, including the effects of neuroplasticity, are not adequately understood [68,71,72]. The results of the current study may have implications for use of TMS in other settings, and the differential TMS activation of inhibitory and excitatory neurons, which was essential to mitigate observed abnormalities of bladder function in patients after SCI within the model, may explain why low frequency TMS leads to reduced cortical excitability and high frequency TMS increased cortical excitability in treating depressive disorders [73]. These issues are not easily approached experimentally, and model-based analyses and predictions may be helpful unraveling these mechanisms and may point to studies that could provide experimental validation.

## 4.5 Limitations of the methods

We modeled the dense network of interneurons throughout the spine as individual nodes, which do not accurately reflect the polysynaptic pathways controlling parasympathetic, sympathetic and somatic control of bladder function [6,18,74]. We ignored afferent information from the sphincter, ignored the diversity of receptors mediating afferent signals, and simplified the efferent control of the sphincter by ignoring the complex parasympathetic, sympathetic and somatic interactions that maintain continence but still allow bladder emptying during detrusor contraction [74]. We modeled the ascending afferent information from both the bladder and the sphincter as direct communication to the PSC when this communication likely occurs through the PAG and is modulated by the action of numerous higher centers in the midbrain and cortex [7,19,24]. We have not accurately reflected many of the changes in neurotransmitter receptors, ion channels and remodeled connections that occur after a SCI [1]. Similar simplifications have been made in previous computational models of bladder control [10–13], and these simplifications are necessary to create a computationally tractable model without so many variables that the model processes are uninterpretable. The simpler representation of bladder control does not, however, diminish the explanatory power of the model; the model is no more complex than it needs to be to provide insight into the possible neural mechanisms whereby TMS ameliorates bladder function after SCI.

Two issues related to neuroplasticity merit further comment. First, the infantile spinal voiding reflex, as embodied in the enhanced model, is fully functional after SCI. The reality is that the spinal voiding reflex emerges slowly after the onset of SCIs above approximately T10. Immediately after SCI, the bladder is areflexic, flaccid, and in a state of urinary retention [18,75]. The spinal voiding reflex wanes in children as a function of development and the emergence of social control over voiding, and once inhibited, we suspect that the synaptic strength of the spinal voiding reflex weakens as the reflex is inhibited and falls into desuetude. This process, at least as studied in kittens, involves the emergence of descending serotonergic

inhibition of the sacral elements of the spinal voiding reflex from the brainstem and a loss of connectivity of afferent pudendal nerves with the effector interneurons and efferent targets of the spinal voiding reflex as the reflex pathway is stimulated less [33]. Hence, immediately after a SCI injury, the process is likely reversed: the descending inhibition of the spinal voiding reflex is removed, but the reflex is weak (after long disuse), but as the afferent and efferent limbs of the reflex are allowed to function, we imagine that the reflex assumes its full strength once the synaptic connections and synaptic strength of the reflex network are re-established over some weeks after the SCI [50]. We did not model this transitional period of restoration of the strength of the spinal voiding reflex, and the results of the enhanced model, therefore, are representative of control of micturition some weeks after a SCI.

The second issue of neuroplasticity is that when TMS was applied serially over the lumbar spine in patients without evidence of volitional control of voiding, some degree of volitional control emerged; the need for regular emptying of the bladder by catheterization was dramatically reduced in some patients [2]. Volitional control of micturition developed in these patients over approximately three weeks of serial, low frequency TMS, and volitional control was lost over 2–3 weeks after TMS was stopped. This neuroplasticity was not incorporated into the enhanced model, and how best to represent this plasticity in the enhanced model must await further research though a TMS-induced neuroplastic loss of inhibition is an appealing mechanism [2,76]. Calcium and the dynamics of post-synaptic calcium changes after different patterns of activation are frequently involved in neuroplasticity. We did not incorporate any calcium dynamics or modifications of synaptic receptors in our models, but both may be targets of future modeling, especially in relation to model plasticity.

The post-SCI emergent spinal voiding reflex is activated by C-fibers rather than A-delta fibers (which conduct much of the other afferent information controlling bladder function). We assumed in our models that all afferent information was equal and similar regardless of receptors generating the information and the axonal velocities of nerves conducting the information. It is reasonable to construct a minimal model based on these assumptions, but it means that all of the observed changes replicated by the model after SCI are due only to changes in interneuron populations, descending signals, and changes in synaptic strength. The foregoing simplifications are justified, for now, in that the simplified models successfully provide a framework within which to develop hypotheses about mechanisms of bladder control, but in no way are these models comprehensive descriptions of the subtle and diverse inputs into the control of micturition. These models represent a starting point for further, more comprehensive model representations of the neural structures underlying the control of micturition. The natural, next steps in elaborating more comprehensive models should include additional sensory and motor control elements that might include, for example, representations of afferent pudendal inputs to bladder control and separate afferent sensory inputs that are differentiated based on receptor and conduction characteristics of A- and C-fiber inputs derived from the bladder.

## 5. Conclusion

We developed a basic framework for model-based analyses of neural circuits controlling micturition after SCI and of the effects of TMS on the spinal cord as a therapeutic intervention to alleviate bladder dysfunction following SCI. A model of the neural circuits was developed by building on a state-of-the-art model to capture the normal storage/voiding functions under healthy conditions and typical disorders of the bladder after SCI [6], such as detrusor-sphincter dyssynergia and overactivity of the bladder. The effect of TMS on the spinal cord was modeled at the single-cell level based on recent experimental findings from patch-clamp [14] and

extracellular recordings [59]. The effect of TMS at the network level was analyzed through model simulations, which revealed possible mechanisms underlying the amelioration of the detrusor-sphincter dyssynergia by TMS and the mechanism whereby low (but not high) frequency TMS was effective for this purpose. Insights from this modeling illustrate how an integrated, systems-level modeling approach can advance the understanding of neural dysfunction, which is essential to determine the neuronal basis of adverse effects of SCI so that rehabilitation and neuromodulation may be used to ameliorate the neurological disorders following SCI.

## Supporting information

**S1 Materials. Table A**: Summary of model equations and terms used in Hodgkin-Huxley formulation of nodes. **Table B** Parameters and variables of Hodgkin-Huxley model. **Table C**: Parameters and variables of network of Hodgkin-Huxley model. **Table D**: Parameters and variable associated with TMS stimulation model. **Table E**: A summary of the axonal (cable) variables, parameters and differential equations used within the model. **Table F**: The nodal parameters and variables used within the model.
(DOCX)

## Author Contributions

**Conceptualization:** Mahshid Fardadi, J. C. Leiter, Daniel C. Lu, Tetsuya Iwasaki.

**Formal analysis:** Mahshid Fardadi, J. C. Leiter.

**Funding acquisition:** Mahshid Fardadi, Daniel C. Lu.

**Investigation:** Mahshid Fardadi, J. C. Leiter.

**Methodology:** Mahshid Fardadi.

**Project administration:** Daniel C. Lu.

**Software:** Mahshid Fardadi.

**Supervision:** J. C. Leiter, Daniel C. Lu, Tetsuya Iwasaki.

**Validation:** Mahshid Fardadi.

**Visualization:** Mahshid Fardadi, J. C. Leiter.

**Writing – original draft:** Mahshid Fardadi, J. C. Leiter.

**Writing – review & editing:** Mahshid Fardadi, J. C. Leiter, Daniel C. Lu, Tetsuya Iwasaki.

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
