## [Decision Letter · Decision Letter 0]

25 Oct 2023

Dear Dr fardadi,

Thank you very much for submitting your manuscript "Model-based Analysis of the Acute Effects of Transcutaneous Magnetic Spinal Cord Stimulation on Micturition after Spinal Cord Injury in Humans" for consideration at PLOS Computational Biology.

As with all papers reviewed by the journal, your manuscript was reviewed by members of the editorial board and by several independent reviewers.

While the reviewers found some element of interest in your paper, it appears that several aspects need substantial improvement.

In light of the reviews (below this email), we would like to invite the resubmission of a significantly-revised version that takes into account the reviewers' comments.

We cannot make any decision about publication until we have seen the revised manuscript and your response to the reviewers' comments. Your revised manuscript is also likely to be sent to reviewers for further evaluation.

Sincerely,

Daniele Marinazzo

Section Editor

PLOS Computational Biology

Reviewer's Responses to Questions

**Comments to the Authors:**

Reviewer #1: This paper describes the use of a computational model to simulate the neural responses to TMS and the resulting bladder reflexes before and after SCI. While an important topic, the information in this paper is sometimes difficult to follow, in part due to inconsistent labeling and references between the text and figures.

Introduction

Page 5.

Clarify the phrase “Epidural stimulation and transcutaneous magnetic…” to be “Epidural electrical stimulation and transcutaneous magnetic…”

In the second paragraph, the first sentence suggests that possible mechanisms for control will be discussed. Instead, existing computational models are discussed, without much justification regarding why these models are useful. I would like to see first a discussion of the key relevant reflexes, then an explanation of why the model presented here will add valuable insights to this field.

Methods/Figure 1.

1A. This figure suggests that the hypogastric nerve does not contain afferents and neglects the bladder neck connectivity of the hypogastric nerve. The “dot” ending of the branch off of the pelvic nerve is unlabeled and unclear; it appears to connect to the ureters in the figure. The green connection to the bladder is unlabeled (is this a limb of the pelvic nerve that has been separated out?) The nodes are colored but there is no key for those colors, and there is no key for the connection colors, which are difficult to see because the lines are very thin. Additionally, it should be noted that these color choices are essentially impossible to distinguish if one is colorblind. I think the best way to address the node issue here is to color them the same color, and number them all; this way they can be referred to consistently between figures 1A and B.

The description of this figure in the text is pretty confusing. The text describes activity at the lumbosacral spinal cord, which is not a region of the spinal cord labeled in the figure. In the text describing of Figure 1A, the authors state: “This bladder-derived afferent information provides a ‘tonic influence’ to activate the PSC, which in turn stimulates neurons in the Nucleus of Onuf that contract muscles of the pelvic floor and the external urethral sphincter. The muscles of the pelvic floor and the external urethral sphincter are also actively contracted during urine storage by efferent activity originating from spinal interneurons that respond to afferent information transmitted from the bladder and urethra through pelvic and pudendal nerves, respectively.” However, the pelvic floor is not labeled in the figure, and the innervation of the pelvic floor is unclear in the literature (in animals, it is innervated by the levator ani nerve; in humans it may be innervated by the levator ani or the pudendal) and the direct involvement of the pelvic floor in LUT reflexes is unclear. Further, activity of the pelvic floor is not directly modeled—it may be unnecessary to mention it. The urethra is not labeled in the figure and it is not clear in the text that activity in the urethra is carried by the pudendal nerve.

Figure 1B, similarly to 1A, would benefit from additional legend elements and simplification of the color scheme. For example, on page 8, a reference is made to the “red circuits, excluding the projection to Node 7.” I was confused by this description – it looks like that’s the color for the pelvic nerve, and node 2 is also red but not touching any red neurons, and there are a couple of red interneurons but they don’t all connect to each other.

Page 8 paragraph 2 -

I’d like a citation for the statement that the lack of PMC descending signals is what causes discoordination of detrusor and EUS.

The framing of the last sentence suggests that overactive bladder is the cause of DSD; rephrase this.

Page 8 paragraph 3 to page 9

By “added lumbar interneurons” – is there a physiological basis for the specific things you added or is it just something arbitrary added to make the model cooperate? Where are they in your figure in terms of node numbers (“shown in blue” is unclear given that there are multiple colors of blue)? What is the justification for each added interneuron? Again, nothing in the figure is labeled “lumbar;” please be consistent with descriptions between the figure and text to keep it clear.

“The other node within the PMC” – please just describe by node numbers and include a figure ref to 1B here.

This is the first mention of social appropriateness, but I don’t see any results relevant to changes in your model at these higher centers; additionally the statement about how social appropriateness is controlled is not cited.

“Bladder filing - indicate” should be “bladder filling, indicating.”

The green arrow described here is a little confusing because there are multiple green connections in the figure.

Page 10

How did you decide on the influence of applied stimulation? Do you know how diffusely TMS spreads? It may activate more than just the interneurons described here, such as direct activation of the pelvic, pudendal, or hypogastric nerves. Specify the frequencies that this stimulation was applied at. Do you expect that TMS would have a frequency-dependent effect through the same mechanisms as nerve stimulation?

Page 11

I am concerned about the description of the changes to the “traditional circuit” described in the second paragraph on page 11 (“The ‘traditional circuit’ has been simplified as shown in Figure 1A in that sphincter afferents in the pudendal nerve are not considered…the effect of pudendal afferent information is, to some extent, redundant with information from the bladder”). While the pelvic afferents are often described (by deGroat and others) as the primary drivers of LUT reflexes, stimulation on the pelvic and pudendal afferents can evoke different responses. The pelvic afferents, when stimulated, tend to produce a bladder contraction which may be accompanied by a sphincter contraction. The pudendal afferents, when stimulated, can produce a frequency-dependent response, as described by McGee (ref 12 in this paper). Describing them as redundant is therefore incomplete, and this description minimizes the potential impact of other neural populations.

Page 13

In the first paragraph, change “seems reasonable” to something like “Acute experiments suggest that this mechanism can approximate…”

Results 3.1/Figure 2.

In the text, the circuit elements in 2A/B are described as “highlighted” but it’s difficult to discern what this includes in the figure, please clarify this.

Figure 2C-D axis labels are hard to read; use consistent font style and size. It’s unclear where these voltages are recorded from – specifically label that it is membrane potential from a certain node for each y axis in C1-D3.

Results 3.2-3.3/Figure 3.

Axis label and figure lettering style has changed from Figure 2, please be consistent. Figure components are misaligned. No y-axis label.

Why do you think C1 shows a rapid firing at the onset of the simulation?

Results 3.4/Figure 4.

The description of the application of TMS to the circuit should likely be in the Methods section, rather than the Results. Labels A-D are in yet a different format compared to previous figures.

Results 3.5/Figure 5.

Is figure 5A-B presenting any new information? It seems to be replicating earlier figures and I don’t see the purple highlight described in the text. Figure 5C is not referred to in the text; the text does refer to a figure 5 E that doesn’t seem to exist so you may have gotten your numbering off by one. Please label axes; lack of labels and mislabeled figure numbers makes it difficult to offer feedback on the results in this section.

Figure 6 appears not to be referenced in the text and the font size is too small to easily read labels.

Discussion

Page 20-21

Two key findings are described: First, that there is an important infantile spinal voiding reflex, and second, that the model must contain excitatory and inhibitory interneurons with different sensitivity. The first seems to me to be less of a significant finding and more of a metric that was used to validate the model, as it is well-established that this reflex exists and is relevant to changes after SCI. The second would be more convincing with additional justification of the reasoning behind the components that were added to this model compared to a “traditional” model.

Given that the post-SCI emergent spinal voiding reflex is activated by C fibers rather than A-delta fibers, is it reasonable to build a model that represents both of those fiber populations as the same and suggests that all of the changes after SCI are due only to interneuron populations and descending signals?

Page 24

Remove the extra parenthesis after “stimulation of the detrusor.”

It’s worth noting that citation 61, here used to describe the downfalls of “afferent or efferent arms of reflex control of the bladder,” primarily discusses tibial and pudendal nerve stimulation, neither of which is considered in this model.

Can you expand on the statement “The differential sensitivity of excitatory and inhibitory interneurons to TMS in the model may explain why low-frequency TMS treatment more effectively restored voluntary control of micturition than 30 Hz TMS?” This statement seems to be one of the key arguments in this paper but I’d like a walkthrough of exactly how the model explains this.

6.1 The site for BioRender should be “BioRender.com” not “BioRenders.com.”

Reviewer #2: review is uploaded as an attachment

**Have the authors made all data and (if applicable) computational code underlying the findings in their manuscript fully available?**

Reviewer #1: Yes

Reviewer #2: **No: **See review attached

PLOS authors have the option to publish the peer review history of their article (what does this mean?). If published, this will include your full peer review and any attached files.

Reviewer #1: No

Reviewer #2: No

Figure Files:

Data Requirements:

Reproducibility:

To enhance the reproducibility of your results, we recommend that you deposit your laboratory protocols in protocols.io, where a protocol can be assigned its own identifier (DOI) such that it can be cited independently in the future. Additionally, PLOS ONE offers an option to publish peer-reviewed clinical study protocols. Read more information on sharing 

---

## [Decision Letter · Decision Letter 1]

12 Mar 2024

Dear Dr fardadi,

Thank you very much for submitting your manuscript "Model-based Analysis of the Acute Effects of Transcutaneous Magnetic Spinal Cord Stimulation on Micturition after Spinal Cord Injury in Humans" for consideration at PLOS Computational Biology.

The effort in producing this revised version, which addressed several critical points, were appreciated.

Still some important aspects still need to be addressed, and we are happy to provide you the opportunity to do so.

We cannot make any decision about publication until we have seen the revised manuscript and your response to the reviewers' comments. Your revised manuscript is also likely to be sent to reviewers for further evaluation.

Sincerely,

Daniele Marinazzo

Section Editor

PLOS Computational Biology

Daniele Marinazzo

Section Editor

PLOS Computational Biology

Reviewer's Responses to Questions

**Comments to the Authors:**

Reviewer #1: This paper covers an interesting model of TMS to activate neural circuits that target the bladder. In this revision, it has been substantially improved by edits to the text and figures. However, some changes, particularly to figures, are necessary to communicate the goals effectively. In particular, it is absolutely essential that the first figure/schematic figure that is repeated throughout the rest of the work is clear to readers in order to communicate the circuits and manipulations.

Comments:

Introduction

>Page 5: After the phrase “Muscle dynamics have been described by Hill equations,” clarify what those are and provide a citation.

>Otherwise, the introduction is now much clearer.

Methods

>Page 7: There is a missing reference error beginning here and for all the Figure numbers.

Figure 1.: 1A. This figure suggests that the hypogastric nerve does not contain afferents and neglects the bladder neck connectivity of the hypogastric nerve. The “dot” ending of the branch off of the pelvic nerve is unlabeled and unclear; it appears to connect to the ureters in the figure. The green connection to the bladder is unlabeled (is this a limb of the pelvic nerve that has been separated out?) The nodes are colored but there is no key for those colors, and there is no key for the connection colors, which are difficult to see because the lines are very thin. Additionally, it should be noted that these color choices are essentially impossible to distinguish if one is colorblind. I think the best way to address the node issue here is to color them the same color, and number them all; this way they can be referred to consistently between figures 1A and B.

Response: Figure 1 was adapted from another publication, and some of the elements requested were not included in the original figure (or model). The original figure emphasizes the efferent control of bladder activity. We are trying to strike a balance between the minimum elements necessary to replicate a behavior within a mathematical model, and the diversity of inputs and outputs that exist in the natural state. In the same spirit, we are trying to provide complete information in the figure, but adding too many elements makes the figure too busy to interpret. Having said that, we revised the figure(s) along the lines suggested by the reviewer, and added test to the legend to make it clear that afferent activity exists in many of the nerves, but it is not shown in the figure. There are no ureters in the figure. The color code is red for urine storage, green for urine emptying and blue if the nerve is involved in both processes (e.g., the hypogastric and pudendal nerves). This is stated in the text. In addition, we added information to the text making it clear that the model we developed has more interneurons than models that went before, but it is still a minimalist representation of the physiology of micturition (which is relevant to comments below as well).

> Reviewer response to response: I understand what you are trying to convey in this figure, but it does not stand alone yet.

>After looking into the Fowler paper you used to generate this figure, I found that the branch from the pelvic nerve with the dot/circle marker in 1A was in fact just the T junction and cell body of the pelvic nerve afferents. However, in this figure, unlike the Fowler figure, it appears to be a separate connection. There are (unlabeled) ureters in the figure (the faded out tubes on the left and right of the bladder image); there is no need to remove them from the figure but the branch from the pelvic nerve that has the dot on the end appears to connect to it as-is. This is misleading; please either include the cell bodies of all cells or remove this one. It is unclear if the various nodes/circles drawn here represent cell bodies (as appears to be the case for ie the hypogastric neurons), brain regions (as appears to be the case for “Higher Centers”), or simply a local area of interest (as appears to be the case for the TMS zone in 1B, since presumably TMS did not spontaneously generate new cell bodies). Additionally, please add a figure legend that includes the color information of each axon. You also appear to be suggesting with this figure that the pelvic afferents (red) contribute to urine storage but not voiding. Work by deGroat and others suggests that this is untrue. Because the shapes of the ends of the neurons are critical for understanding the figure, please make sure they are not overlapping with each other or with the circles you have drawn.

>In the caption, I would recommend beginning with a sentence such as “A. The circuits for micturition and continence are both shown in this figure.” to clarify the overlapping nature. You also contradict yourself in this caption: you say that the only afferents that are relevant are pelvic, but you have drawn the pudendal neuron with the ending connection labeled as afferent in the key. Can you please additionally explain why the nodes/cell bodies/circles in the figure are colored differently and what those colors mean? I don’t see that information for all of the nodes in the figure itself or in the caption.

>Figure 1B – many of the items from the notes on 1A above would also improve 1B. In particular, overlapping elements make it difficult to follow some of the paths. I would add that I don’t understand how the pelvic nerve branches as shown here, or why there are suddenly two PMC nodes. You show the hypogastric activity emerging from the lumbar spine here and the thoracolumbar spine in 1A, please pick one and give a reference for the origin of the hypogastric efferents in the cord.

>Regarding the equations in this section, I would agree with the other reviewer that a table format could make it clearer what each term is and what the purpose of each equation is.

>Figure 2 (and remaining figures) – please make the letters indicating figure parts all black and remove the parentheses. Please include a key in the figure itself that shows the meaning of the different colored lines (I know it is in the caption, but it will be much easier to read with a key). I assume that the black, red, and blue lines have different y axes? If so, please indicate in some way what their starting potentials are or where zero is for each of the traces.

> Figure 3 – I appreciate the highlighted thicker lines, but it is confusing that it is a different color than the original lines. Can you make the lines thicker in their original colors instead, or if you need these to be highlighted in a certain color, can you remove the thinner line or make it black/a neutral color? What is it that happens around 1800 ms that changes the behavior of so many of these nodes? Given that this is the figure referred to for the “Healthy” section, I would prefer that it doesn’t have the dotted line on the figure indicating an SCI.

Discussion

>It may be helpful, given the opening framing of the discussion, to provide a table with the differences between the traditional model and the novel model, and the reasoning for each addition. This would be easy to refer back to rather than needing to go through text line-by-line.

> I am glad you added a note on the limitation of different fiber types on page 22, but if someone isn’t aware of the different roles of different fiber types in control of the bladder they may not have enough information here. I’d like to see a comment (just one sentence) that specifically addresses the differing roles of A-delta and C fibers in control of the bladder before/after SCI to give context.

>Could you explain further and give some physiological data for the statement “prolonged bursting in the model that is likely achieved in actual neuronal tissue by networks of neurons”?

Reviewer #2: The authors have responded adequately to the queries, made relevant changes to their manuscript, and provided requisite clarifications.

The responses are acceptable. Apropos the larger point made regarding the sparsity of model parameters being extracted from experimental literature/work specific to the bladder, it is important to note that while bladder-specific data might be sparse in relation to the components of the circuitry modelled by the authors, it is nevertheless useful and instructive to construct frameworks such as the one introduced here, to derive insights into bladder control and function. Model parameter values are within physiological bounds and are typical for individual components of the cellular machinery. The predictions and analyses presented are therefore relevant.

Considering also the relevance of the work to the field, I recommend acceptance of the paper for publication.

**Have the authors made all data and (if applicable) computational code underlying the findings in their manuscript fully available?**

Reviewer #1: Yes

Reviewer #2: Yes

PLOS authors have the option to publish the peer review history of their article (what does this mean?). If published, this will include your full peer review and any attached files.

Reviewer #1: No

Reviewer #2: No
---

## [Decision Letter · Decision Letter 2]

7 Jun 2024

Dear Dr fardadi,

We are pleased to inform you that your manuscript 'Model-based Analysis of the Acute Effects of Transcutaneous Magnetic Spinal Cord Stimulation on Micturition after Spinal Cord Injury in Humans' has been provisionally accepted for publication in PLOS Computational Biology.

Best regards,

Daniele Marinazzo

Section Editor

PLOS Computational Biology

Daniele Marinazzo

Section Editor

PLOS Computational Biology

Reviewer's Responses to Questions

**Comments to the Authors:**

Reviewer #1: This is an interesting and valuable study and the authors have addressed my concerns.

**Have the authors made all data and (if applicable) computational code underlying the findings in their manuscript fully available?**

Reviewer #1: Yes

PLOS authors have the option to publish the peer review history of their article (what does this mean?). If published, this will include your full peer review and any attached files.

Reviewer #1: No

---

## [Editor Report · Acceptance letter]

25 Jun 2024

PCOMPBIOL-D-23-01456R2 

Model-based Analysis of the Acute Effects of Transcutaneous Magnetic Spinal Cord Stimulation on Micturition after Spinal Cord Injury in Humans

Dear Dr fardadi,

I am pleased to inform you that your manuscript has been formally accepted for publication in PLOS Computational Biology. Your manuscript is now with our production department and you will be notified of the publication date in due course.

With kind regards,

Zsofia Freund
